# GIP/GLP-1RA as adjunctive to automated insulin delivery in adults with Type 1 diabetes (the AID-JUNCT trial): Study protocol for a prospective, randomized, clinical trial

Maria Carolina Fragozo-Ramos[1,2], Gabriela Schenker[3], Matthias Hepprich[4], Jaime Gallo-Villegas[5,6], Thomas Züger[1,3‡], Jose Garcia-Tirado[1,2‡]*

1 Department of Diabetes, Endocrinology, Nutritional Medicine and Metabolism, Inselspital, Bern University Hospital and University of Bern, Bern, Switzerland, 2 Diabetes Center Berne, Bern, Switzerland, 3 Department of Endocrinology and Metabolic Diseases, Kantonsspital Olten, Olten, Switzerland, 4 University of Basel, Medical Faculty, Basel, Switzerland, 5 Faculty of Medicine, Sports Medicine Postgraduate Program and GRINMADE Research Group, University of Antioquia, Medellín, Colombia, 6 SICOR Center, Medellín, Colombia

‡ TZ and JGT are Joint Senior Authors.
* jose.garcia@unibe.ch

## Abstract

### Background

Glycemic control in type 1 diabetes (T1D) remains a challenge, with 20−30% of adults achieving an A1c target of <7%. Glucagon-like peptide 1 receptor agonist (GLP-1 RA) and dual glucose-dependent insulinotropic polypeptide (GIP) and GLP-1 RA (GIP/GLP-1 RA) have emerged as a promising therapy in T1D. Previous studies have shown that patients with T1D can significantly improve glycemic control while experiencing a reduction in insulin dose and body weight when long-acting GLP-1RAs or GIP/GLP-1RAs are added to insulin therapy. However, randomized controlled trials (RCT) are still insufficient.

### Methods

This is a prospective, randomized, parallel-group, open-label, superiority-controlled design that evaluates the safety and efficacy of adding tirzepatide to insulin therapy in participants with T1D under automated insulin delivery (AID) control. We will enroll 42 participants aged 18–65 years with confirmed T1D diagnosis ≥12 months, currently on AID insulin therapy for at least three months, with A1C ≥ 6.5% and ≤ 10%, and BMI ≥ 23 kg/m². Participants will be randomized in a 1:1 ratio to either tirzepatide with a target dosage of 5.0 mg (after titration) or standard of care (SoC) for 16 weeks. The primary endpoint is continuous glucose monitoring (CGM)-measured percent time spent between 3.9 and 10.0 mmol/L (TIR) from baseline to follow-up after 16 weeks of treatment. Secondary endpoints include: the CGM-measured change in

**Data availability statement:** No datasets were generated or analyzed during the current study. All relevant data from this study will be made available upon study completion.

**Funding:** This work was supported by internal seed grant funds from the University of Bern awarded to Jose Garcia-Tirado. Dexcom provided sensors at a research discount for this study.

**Competing interests:** This work was supported by internal seed grant funds from the University of Bern awarded to Jose Garcia-Tirado. Dexcom provided sensors at a research discount for this study. There are no patents, products in development, or marketed products associated with this research to declare. This does not alter our adherence to PLOS ONE policies on sharing data and materials.

24/7 percent time >10.0 mmol/L, > 13.9 mmol/L, < 3.9 mmol/L, < 3.0 mmol/L. The exploratory endpoints include: the change in body mass index (BMI), liver steatosis (MASLD), and body composition. Safety outcomes include severe hypoglycemia, diabetic ketoacidosis (DKA), and refractory gastrointestinal side effects.

## Discussion

This is the first prospective study to investigate the safety and efficacy of tirzepatide (GIP/GLP1-RAs) as an adjuvant therapy to AID in T1D. This study may contribute unique data to significantly improving glucose and cardio-metabolic outcomes, re-directing attention to further treatment in T1D beyond insulin therapies.

## Introduction

### Background and rationale

Type 1 diabetes (T1D) is an autoimmune condition resulting in an insulin deficiency and a lifelong need for insulin replacement [1]. Achieving optimal glycemic control in T1D remains challenging, with only about 20–30% of adults reaching a hemoglobin A1c (HbA1c) target of <7% [2,3]. This is despite the availability of modern insulin analogs [4], and the growing acceptance of continuous glucose monitoring (CGM) [5] and AID systems [6]. These emerging technologies have demonstrated clear benefits in improving glycemic outcomes and reducing hypoglycemia [7]. However, excess mortality, weight gain, and cardio-metabolic complications remain higher in people with T1D [8,9].

Recent studies have documented the rising prevalence of overweight (OW) and obesity (Ob) among individuals with T1D [10,11]. Many of these individuals now meet the criteria for metabolic syndrome (MS), often presenting with higher HbA1c levels, elevated insulin requirements, and cardio-metabolic complications such as metabolic dysfunction-associated steatotic liver disease (MASLD), and renal dysfunction [12–14]. Moreover, epidemiological data reveal that individuals with T1D have significantly reduced life expectancies, approximately 11–13 years shorter [15] and experience cardiovascular disease (CVD) events, on average, more than a decade earlier than the general population [16]. Alarmingly, CVD has been consistently identified as the leading cause of death in T1D [17–19], with studies showing that T1D confers a substantial risk of CVD even when conventional treatment targets are achieved [20].

GLP-1 receptor agonists (GLP-1RAs) and dual incretin analogs (GIP/GLP-1RAs) have emerged as promising therapies for T1D [21–23]. These medications reduce plasma glucose levels with a non-significant risk of hypoglycemia or diabetic ketoacidosis (DKA) [24,25]. The primary mechanisms responsible for these effects include (i) glucose-dependent insulinotropic action [in subjects with residual β-cell function], (ii) suppression of glucagon hypersecretion, except during hypoglycemic episodes, and (iii) delayed gastric emptying, all of which help to reduce postprandial glycemic excursions [26]. Additionally, GLP-1RAs and GIP/GLP-1RAs induce early, acute changes in appetite and satiety, leading to reduced caloric intake and weight loss [27,28].

In T2D, GLP-1RAs and GIP/GLP-1RAs have demonstrated significant reductions in major adverse cardiovascular events (MACE) and renal outcomes in several large clinical studies [29–33]. Additionally, GIP/GLP-1RAs have shown preliminary efficacy in treating metabolic dysfunction–associated steatohepatitis (MASH) [34]. However, in T1D, no current treatment modality addresses both glycemic control and cardio-metabolic outcomes, even though only a minority of patients achieve glucose targets, and this population faces markedly increased metabolic disturbances and cardiovascular risk [2,13,16,35].

While GLP-1RAs and GIP/GLP-1RAs show potential for improving glucose control and reducing cardiovascular risk in T1D, current clinical evidence is insufficient to recommend their routine use in this population. To address this gap, we plan to conduct a prospective, randomized, parallel-group, open-label, superiority-controlled clinical trial to test the hypothesis that combining a GIP/GLP-1RA (tirzepatide) with available AID systems is superior to AID alone on glucose control in individuals with T1D. The primary aim of this study is to evaluate the efficacy and safety of tirzepatide as an adjunct to AID therapy.

## Materials and methods

### Trial design

We will conduct a prospective, randomized, parallel-group, open-label, superiority-controlled clinical trial. The study has been designed according to Standard Protocol Items: Recommendations for Intervention Trials-SPIRIT guidelines [36] (**S1 and Fig 1**). The overall study design and participant flow are illustrated in **Figs 2** and **3**, respectively.

This study will be conducted on individuals living with T1D. Only people with T1D using an AID system approved by Swissmedic (The Swiss Agency for Therapeutic Products) will be included. No vulnerable populations, such as neonates, pregnant women, children, prisoners, institutionalized individuals, or others who may be considered part of a vulnerable population, will be included in this study. The target population will be adults treated and referred to the Department of Endocrinology, Diabetes, Nutrition, and Metabolism at Kantonsspital Olten (KSO). Eligibility will be assessed based on exclusion and inclusion criteria.

Inclusion criteria:

- Diagnosed with T1D for at least 12 months, aged 18–65 years.

- Currently on AID therapy for at least 3 months.

- HbA1c between 6.5% and 10%, and BMI ≥ 23 kg/m2.

- Willing to use tirzepatide once weekly for 16 weeks and wear the study sensor, Dexcom G7 (Dexcom, Inc., San Diego, CA).

- Agree to refrain from starting new non-insulin glucose-lowering agents (e.g., metformin, DPP-4 inhibitors, SGLT2 inhibitors) during the trial.

- Have stable weight (± 5%) for at least 90 days before screening and agree not to initiate weight loss programs.

- Females of childbearing potential and males must commit to the use of reliable contraception.

- Willing to provide signed informed consent and follow the protocol.

Exclusion criteria:

- Recent history of diabetic ketoacidosis or severe hypoglycemia.

- Uncontrolled diabetic retinopathy, severe gastroparesis, or insulin treatment for less than 12 months.

- Estimated glomerular filtration rate (eGFR) below 30 mL/min/1.73 m2, pregnancy, breastfeeding, or intention to become pregnant.

| | Enrollment | Allocation | Post-Allocation | | | | Close-out |
|---|---|---|---|---|---|---|---|
| **TIME POINT** | **V1** | **V2** | **P1** | **V3** | **P2** | **V4** | **P3a** |
| Weeks | -2 | 0 | 4 | 8 | 12 | 16 | 18 |
| Window (days) | -7 | 14 up to 21 days after V1 | ±3 | ±3 | ±3 | ±3 | ±7 |
| **ENROLLMENT:** | | | | | | | |
| Informed consent | X | | | | | | |
| Inclusion/Exclusion | X | | | | | | |
| Randomization | | X | | | | | |
| **INTERVENTION:** | | | | | | | |
| Intervention A: GIP/GLP1RA | | ⟶ | | | | | |
| Intervention B: SoC | | ⟶ | | | | | |
| **ASSESMENTS** | | | | | | | |
| Weight, Height, Waist & Hip circumference | X | | | | | | |
| Vital signs (BP, HR, Temperature[c]) | X | X | | X | | X | |
| ECG | X | | | X | | X | |
| Body composition (BIA) | | X | | | | X | |
| Hepatic elastography (*FibroScan®*) | | X | | | | X | |
| Device downloads (CGM and pump data) | | X | X | X | X | X | |
| SNAQ app data downloads | | X | | X | | X | |
| Activity tracker downloads | | X | | X | | X | |
| Questionnaires (DTSQs-c) | | X | | | | X | |
| HbA1c (central) | X | | | | | X | |
| Hematology (hb-hct) | X | | | | | | |
| Serum/urine pregnancy test (for child-bearing potential women)[d] | X | X | | X | | X | |
| Hepatic chemistry/coagulation | X[e] | | | X[f] | | X[f] | |
| Endocrine | X[g] | | | | | X[h] | |
| Pancreas (exocrine) panel | X | | | X | | X | |
| Lipid panel (fasting) | X | | | | | X | |
| Glucose fasting | X | | | | | | |
| Renal chemistry | X[i] | | | X[j] | | X[i] | |
| AE/SAE assessment, including assessment for clinical significance and severe hypoglycemia, and DKA | X | X | X | X | X | X | X |

**Fig 1. SPIRIT Schedule of Enrolment, Interventions, and Assessments.** Abbreviations: BMI = body mass index; BG = blood glucose; BP = blood pressure; HR = heart rate; BIA = bioelectrical impedance analysis; CGM = continuous glucose monitoring; DTSQs-c = diabetes treatment satisfaction questionnaire status and change versions; ECG = electrocardiogram; U/S = unscheduled; P/D = premature discontinuation; DKA = diabetes ketoacidosis; HbA1c = hemoglobin A1c; Hb = hemoglobin; HCT = hematocrit; IMP = investigational medicinal product (study drug); FDA = Food and Drug Administration. [a]Only for the participants in the intervention arm. [b]Included: gender, race/ethnicity, and year of birth. [c]The temperature should be measured at the screening visit (V1) and randomization visit (V2) to exclude infection. [d]A serum pregnancy test will be performed at the screening visit, the result must be available before randomization and the first injection of the study drug for women of childbearing potential only. Additional pregnancy in urine tests will be repeated during the study [e]Include: total bilirubin, direct bilirubin, alkaline phosphatase (ALP), partial thromboplastin time (PTT), prothrombin time (PT), international normalized ratio (INR), alanine aminotransferase (ALT), aspartate aminotransferase (AST), and gamma-glutamyl transferase (GGT). [f]Include: total bilirubin, direct bilirubin, ALT, AST. [g]Include: thyroid-stimulating hormone (TSH), free thyroxine (FT4), calcitonin, C-peptide (fasting), estradiol, and follicle-stimulating hormone (FSH). [h]Only calcitonin. [i]Include: creatinine (with eGFR by CKD-EPI), and urine microalbumin/creatinine ratio (uACR). [j]Only creatinine (with eGFR by CKD-EPI).

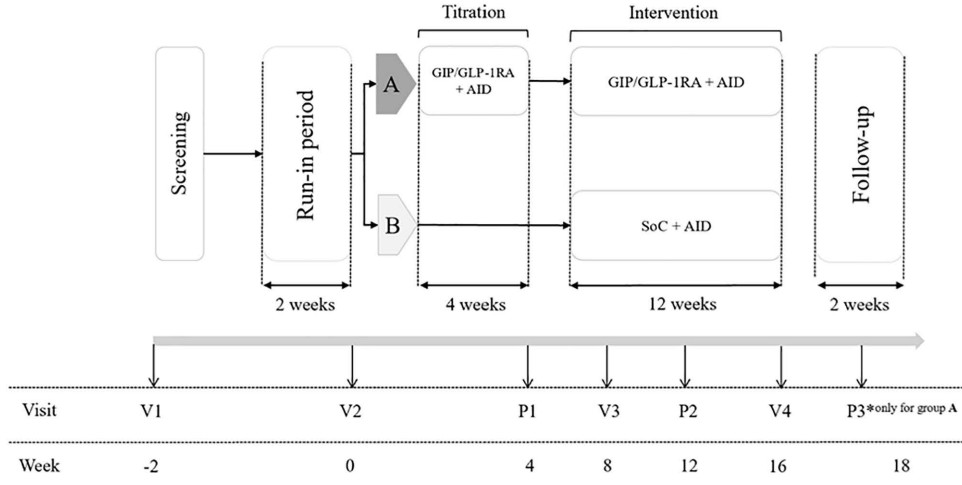

**Fig 2. General Study Design.** Schematic representation of the overall study design, including run-in period, randomization, intervention, and follow-up periods.

- Active or uncontrolled cardiovascular, hypertension, or seizure disorders.

- History of allergies to GLP-1RAs/GIP-RAs, multiple endocrine neoplasia type 2A or 2B (MEN2A or MEN2B), medullary thyroid carcinoma, or malignancy (except specific types of skin, cervix, or prostate cancer).

- Severe liver or pancreatic conditions, cystic fibrosis, gastric bypass, or uncontrolled thyroid disease.

- Recent use of weight loss medications, chronic glucocorticoids, or drugs affecting glucose metabolism.

- Severe psychiatric conditions, including major depression or uncontrolled drug/alcohol abuse.

- Participation in another clinical trial, investigational drug use, or enrollment of investigator family members.

- Other significant active medical conditions affecting participation or data interpretation.

Participants will receive an information sheet and consent form containing detailed study information (**S2 File**). Formal consent will be obtained using the approved form before any investigative or study-related procedures are carried out. The informed consent or assent process will be conducted by the principal investigator or a designated delegate. The study physician will also assess the eligibility criteria for each participant and determine whether they should be included in the trial.

## Intervention

Arm "A" (Intervention Group) will receive tirzepatide for 16 weeks. A maximum dose of 5 mg SC weekly will be administered as a target dose (after the titration phase, starting with 2.5 mg SC/week per four weeks). All participants will receive verbal and written education on tirzepatide pen use.

Arm "B" (Control Group) will continue with the Standard of Care (SoC) and concomitant medication (if not part of the exclusion criteria). Participants in the control group will continue with their home treatment according to international guidelines and the criteria of their treating physician [1].

**Dose modifications.** In this study, participants assigned to arm A will begin with an initial dose of 2.5 mg of tirzepatide once weekly. After four weeks, the dose will be increased to a stable dose of 5 mg once weekly. In cases of medication

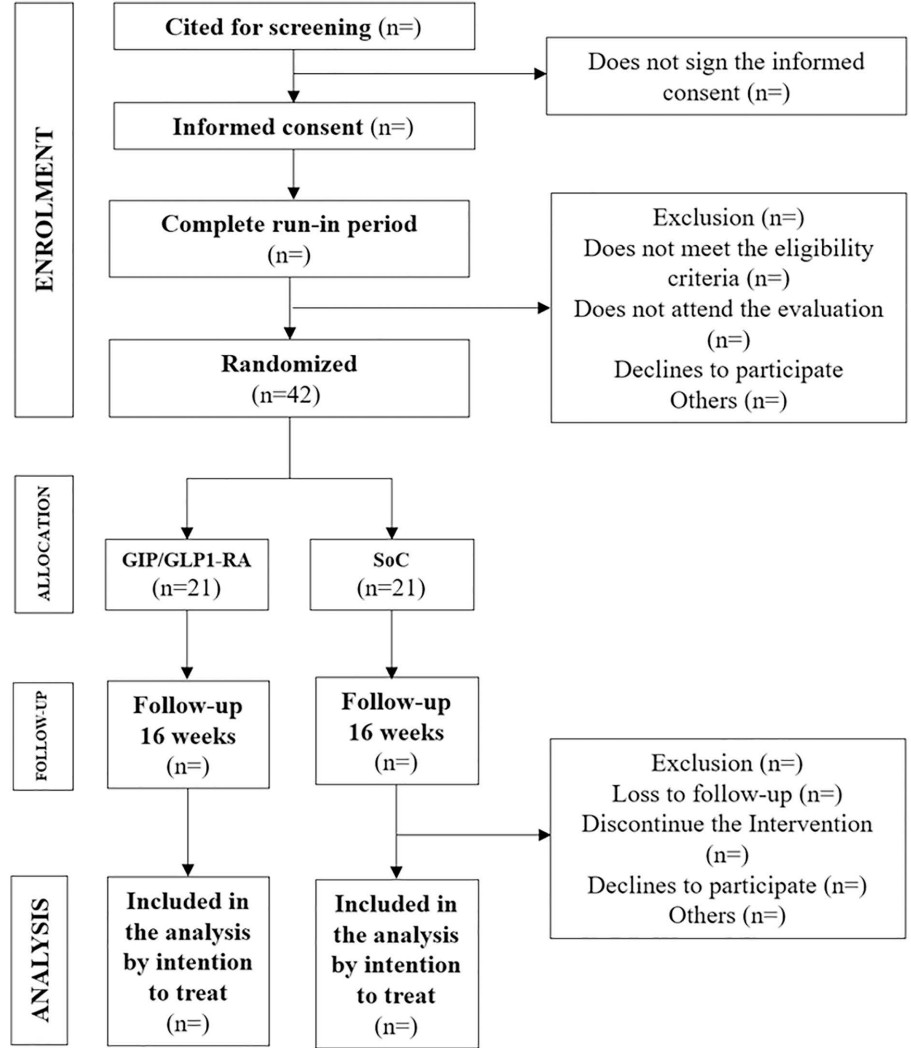

**Fig 3. Study Flowchart.** Flowchart of participant progression through the trial, showing screening, randomization, allocation, follow-up, and analysis.

intolerance, the dose may be reduced. During the dose escalation period, investigators should make every effort to titrate and maintain participants on the target tirzepatide dose of 5 mg/week. Once the stabilization phase is complete, no further dose adjustments will be permitted unless a safety concern or intolerance arises.

If a dose is missed, participants will be instructed to administer tirzepatide as soon as possible, within four days (96 hours) of the missed dose. If more than four days have passed, the missed dose should be skipped, and the next dose should be administered on the regularly scheduled day. If necessary, participants may adjust their weekly administration day provided that at least three days (72 hours) separate two doses. The investigator may also discontinue the medication if safety concerns arise, as assessed through the safety outcomes. Additionally, for participants with a normal BMI at baseline, the medication can be reduced to 2.5 mg or discontinued if weight loss exceeds 5% of total body weight or if weight reduction occurs too rapidly (defined as more than 1% of total body weight per week) following a clinical evaluation and based on clinical criteria.

**Compliance with the study intervention.** Tirzepatide compliance will be assessed through weekly entries recorded by participants in a logbook, which the investigator will review during each on-site study visit. Participants will be asked to complete any missing entries as needed. Missed doses will be documented as 'not taken', and participants will be deemed non-compliant if the investigator judges them to have intentionally or repeatedly taken less than the prescribed medication. Compliance with other study procedures will also be evaluated at each visit based on adherence to the visit schedule, completion of the logbook, and any other parameters deemed necessary by the investigator. Participants showing poor compliance with the medication or study procedures will receive additional training and reminders about the importance of adherence. At the end of the study, compliance will be defined as taking at least 75% of the required doses of the study drug.

## Outcomes

**Primary outcome.** The primary endpoint is the mean difference in the CGM-measured percentage of time spent between 3.9 and 10 mmol/L (Time in range; TIR) between groups at week 16. This is a clinically meaningful CGM-based metric, strongly correlated with HbA1c and the risk of long-term diabetes complications [37,38].

**Secondary outcomes.** Secondary outcomes include the mean difference in total daily insulin (TDI), HbA1c, TIR from 07:00–23:00 ($TIR_{7-23}$), time spent between 3.9–7.8 mmol/L (Time in tight range; TITR), and CGM-measured time spent above 10.0 mmol/L (Time above range; TAR), above 13.9 mmol/L (Time in very high range, TVHR), below 3.9 mmol/L (Time below range, TBR), and below 3.0 mmol/L (Time in very low range; TVLR) between groups at week 16.

**Exploratory outcomes.** Exploratory outcomes include the mean difference in the CGM-based metrics: $TIR_{7-23}$, TAR, TVHR, TBR, TVLR, and glucose variability (coefficient of variation) between groups at weeks 8, 12, and 16. Additional exploratory measures include TDI and Total Daily Bolus (TDBo) between groups at weeks 8, 12, and 16.

At week 16, we will also assess mean differences between groups in body weight, waist/hip circumference, waist-to-hip ratio, body fat percentage, fat mass index (FMI), visceral adipose tissue (VAT), lean mass index (LMI), and appendicular lean mass index (ALMI). Furthermore, lipid profiles (LDL, HDL, non-HDL cholesterol, triglycerides, [TG]), TG-glucose index, systolic/diastolic blood pressure, urine albumin-to-creatinine ratio (uACR), hepatic steatosis, and MASLD biomarkers will be assessed.

**Additional outcomes of interest.** These include reported eating behavior and food intake (using the SNAQ app) [39] and patient-reported outcomes via the Diabetes Treatment Satisfaction Questionnaire (DTSQ) [40] at baseline and the end of treatment.

**Safety outcomes.** Safety outcomes (adverse events [AE] and serious adverse events [SAE]) will include clinically significant and severe hypoglycemia, rescue therapy for hyperglycemia or diabetic ketoacidosis, refractory gastrointestinal side effects, dehydration, acute renal events, adjudicated hepatic or pancreatic adverse events, allergic reactions, heart rate changes, treatment-emergent adverse events (TEAEs), and early drug discontinuation due to AE.

## Participant timeline

**Screening.** At Visit 1 (Screening), after written informed consent is given, study staff will assess eligibility criteria, including demographics, medical history (diabetes, hypoglycemia, DKA, complications), and substance use. A physical exam will be performed, including measurements of weight, height, waist/hip ratio, and vital signs, along with an electrocardiogram (ECG) and serum pregnancy test for women of childbearing potential. Laboratory tests will include a variety of blood and urine markers, such as liver enzymes, glucose, HbA1c, and kidney function. The participant will receive a hypoglycemia diary and diabetes management information. Participants will be provided with study devices, including a study phone, activity tracker, ketone meter, and CGM system, and trained to use study apps for data collection.

**Run-in period.** A run-in period will be conducted to gather baseline data before randomization and assess adherence to the study protocol. This period will last between 14 and 21 days. Participants will continue their standard insulin treatment and use their home insulin pump and a study-provided Dexcom G7 sensor (Dexcom; San Diego, CA). They will also track their dietary history using the SNAQ app [39] during the five days before the next visit. Interim visits or phone contacts may be scheduled to address any issues, including assessments of insulin treatment compliance, skin reactions from CGM use, and eligibility for continued participation. To be eligible for randomization, participants must collect data for at least 70% of the run-in phase. If this criterion is not met, the run-in period may be extended at the investigator's discretion.

**Randomization and follow-up visits.** At Visit 2 (Randomization), participants will be randomly assigned to one of the two groups: (A) tirzepatide+AID arm or (B) SoC+AID arm. In the tirzepatide group, participants will begin with a 2.5 mg weekly dose for four weeks, followed by an up-titration to 5.0 mg weekly for 12 weeks (16 treatment weeks). Both groups will undergo assessments of vital signs, body composition, MASLD, and liver fibrosis, and complete the DTSQs [40]. During the study visits, device data from the CGM and insulin pumps will be downloaded. Phone visits will be scheduled for data review, insulin pump adjustments, and adverse event monitoring during the follow-up phase.

According to the study schedule, two additional clinical visits will occur (**Figs 1** and **2**). Visit 3 will include follow-up assessments, such as weight, waist/hip ratio, laboratory tests, and a medication compliance review. At Visit 4 (the end of the intervention, at week 16), final outcome assessments will be conducted, and all study devices will be returned. A follow-up phone visit will be scheduled for the intervention group to evaluate safety outcomes after the cessation of Tirzepatide.

In the event of the study drug's premature discontinuation (PD), an additional visit may be conducted. The outcome assessments for this PD visit will mirror those of Visit 4 (end of intervention). A phone follow-up will be conducted two weeks after the PD visit to monitor safety outcomes (AE/SAE assessment). Additionally, unscheduled visits may be performed at any point during the study per the participant's request or at the investigator's discretion. The assessments for these visits will be guided by clinical judgment.

## Sample size

Assuming a 1:1 randomization ratio, we calculated that a sample size of 42 participants (21 in each arm) would provide 80% power with a type I error rate (two-sided) of 5% to reject the null hypothesis with the following assumption: a standard deviation (SD) at baseline for TIR of 15%, a pre-post difference of TIR in the control group of 0.2%, a pre-post difference of TIR in the treatment group of 10.5%, a correlation of 0.7, and an attrition rate of 20%, which is typical in therapeutic clinical trials [41,42]. The assumptions were taken from retrospective studies in T1D using tirzepatide [22,23].

Participants will be recruited at the investigator's clinic or by referral from healthcare professionals from clinics, hospitals, and practices. In parallel, posters and flyers will be strategically placed in high-traffic areas, such as waiting rooms of hospitals, clinics, and pharmacies, where potential participants or their caregivers will likely see them. We also plan to advertise the study through local newsletters and geo-targeted social media (e.g., LinkedIn, Facebook, Instagram, etc.) and in diabetes-focused groups and organizations to ensure better reachability.

Interested people can connect directly with the study team to express their interest in participating. Motivated individuals will be invited to a screening visit. During this visit, open questions related to the study will be clarified, and after signing the informed consent form, eligibility will be assessed.

## Randomization

Randomization will take place at visit 2. Participants will be allocated using the minimization method (Minimpy v0.3) with the following balancing factors: BMI (<27 and ≥27 kg/m2) and HbA1C (<7.5 and ≥7.5%). To include the next participant, the 'biased coin' method will be used with a base probability of 0.7, and variance will serve as a measure of imbalance

between the groups [43,44]. Two external researchers will manage the participant randomization at a ratio of 1:1 (tirzepati-de+AID, n = 21; SoC + AID, n = 21). The communication between the researchers carrying out the intervention and those tasked with assigning treatment groups will be conducted through email to maintain independence.

Due to the nature of the study intervention and design, neither the participants, the individuals performing the intervention, nor the treating clinicians will be blinded. The main statistical analysis to evaluate the hypothesis will be masked to reduce the bias.

### Data collection methods

**CGM-based and AID metrics.** Primary, secondary, and exploratory outcomes based on CGM metrics and AID systems will be collected and evaluated using the study CGM and the participant's AID system from baseline to weeks 4, 8, and 16 of escalated treatment. To standardize glucose data collection independently of the AID system, all participants will receive Dexcom G7 CGMs, regardless of their home therapy, unless they already use a G7. Data from the study sensor will not be blinded and used solely for research purposes. The G7 CGM sensor should be placed on the arm or abdomen and replaced at least once every ten days by the manufacturer's instructions [45,46].

**Clinical history and physical evaluation.** A complete clinical history, including sociodemographic information, habits, and personal and family histories, will be fulfilled. The anthropometric measurements and records of the vital signs (blood pressure and heart rate, measured by pulse) will be performed according to the protocols for the World Health Organization's STEPwise approach to Surveillance (STEPS) (WHO 2017) [47]. Height and body weight will be measured with Seca Digital Ultrasonic Measuring Rod® (Seca, Germany), with an accuracy of 0.1 kg. Waist circumference (WC) will be measured with a fiberglass anthropometric tape at the intermediate point between the lower edge of the last rib and the iliac crest in the horizontal plane (approximately 2.54 cm above the navel). The hip circumference will be measured around the broadest part of the buttocks, ensuring that the tape is snug but not tight and parallel to the floor. Body mass index (BMI) will be calculated with the formula weight/(height)$^2$ in kg/m$^2$.

Vital signs will be measured, followed by blood sample collection for laboratory testing. Participants should sit quietly for five minutes before vital sign measurements. For each parameter, two measurements will be taken using the same arm, preferably the nondominant arm. The recordings should be taken at least one minute apart. Each pulse and blood pressure measurement will be performed sitting and recorded in the electronic Case Report Form (eCRF). Blood pressure will be taken with an automated blood pressure instrument.

For each participant, a 12-lead ECG will be collected. The electrocardiograms will be recorded after the participant has been supine for five minutes in a quiet room. Electrocardiograms will be interpreted by the investigator or a qualified designee at the study site as soon after ECG collection, ideally when the participant is still present for immediate action if any clinically relevant findings are identified. Any clinically significant findings from ECGs that result in a diagnosis after the participant receives the first dose of the study drug will be reported as an AE via the eCRF. Collected ECGs will be archived as a paper record to preserve the source data.

**Body composition.** Bioelectrical impedance (BIA) analysis will assess global and regional fat mass (*Seca mBCA 514®*). The BIA has an accuracy comparable to the gold standard when performed under standardized conditions [48]. For this study, fat mass percentage (FM%), fat mass index kg/m$^2$ (FMI [total FM/height$^2$]), visceral adipose tissue liter ([l] VAT), lean mass index kg/m$^2$ (LMI [total LM/height$^2$]) and appendicular lean mass index (ALMI [total LM/height$^2$]) will be considered relevant because they have been associated with an elevated risk of cardiometabolic diseases and mortality [49,50].

**Metabolic dysfunction-associated steatotic liver disease (MASLD).** MASLD and liver fibrosis will be assessed by vibration-controlled transient elastography (VCTE) utilizing *FibroScan®* (Echosens, Paris, France). Liver stiffness measurement (LSM) and controlled attenuation parameter (CAP) will be evaluated. The LSM is expressed in kilo pascals (kPa) that correlate with the fibrosis stage, and the CAP value is expressed in decibels per meter that correlates with liver

steatosis grade [51]. These values will be correlated with MASLD biomarkers and the physical evaluation to calculate the hepatic steatosis index (HSI) and fibrosis score (FIB-4).

**Biochemical evaluation.** The laboratory analysis to assess the HbA1c, estimated glomerular filtration rate (eGFR), urinary albumin-to-creatinine ratio (uACR), liver enzymes, lipids, and others will be conducted according to the study schedule (**Fig 1**). The samples will be processed in the study site's central laboratory and reported in the International System of Units (the central laboratory reference ranges will determine the high and low laboratory limits). The central laboratory will use the CKD-EPI equation to estimate and report eGFR [52].

**Diabetes satisfaction and eating behavior.** The overall diabetes satisfaction will be assessed through the Diabetes Treatment Satisfaction Questionnaires (DTSQ), status, and change [40].These questionnaires are for evaluating thoughts, concerns, and distress related to diabetes and general quality of life assessments. The status version (DTSQs) will be self-applied to the participants in paper form at baseline, and the change version (DTSQc) at the study's last visit. The study team will record the information provided in the corresponding eCRF.

Patient-reported eating behavior and food intake will be assessed with data collected from the SNAQ App [39], where participants will be asked to record their food intake for five days before each on-site study visit (**Fig 1**).

Participants will use the Fitbit Sense 2 (Fitbit, Inc., San Francisco, CA) activity tracker during the up-titration and assessment phases. The study will utilize iPhone or Android smartphones compatible with the study devices. Participants will be provided with a smartphone pre-installed with the necessary apps to connect to the wearable devices and will receive instructions on their use; alternatively, they may choose to use their own compatible device. To protect participants' identities, pseudonyms will be assigned for all study-specific accounts.

Data collection will be conducted using REDCap (Research Electronic Data Capture; Vanderbilt University) in the corresponding eCRF [53]. Each participant will be assigned a unique identification number to ensure proper coding and identification of all records.

De-identified data from AID systems and CGMs will also be uploaded into the corresponding eCRF. Investigators at the study site will receive training in data collection procedures as needed.

A Clinical Data Monitor (CDMon) will frequently oversee the data collection flow in REDCap to ensure data quality, completeness, and consistency.

To promote participant retention and ensure complete follow-up, we will provide detailed information about study procedures, benefits, and expectations through informed consent (**S2 File**). Frequent phone calls or email communication will be offered and maintained as needed. A dedicated study coordinator will be the primary point of contact to address participants' concerns and provide support. Travel costs will be covered, and stipends will be provided after each completed visit.

## Data management

**Data management system.** REDCap (Research Electronic Data Capture; Vanderbilt University) will be our clinical data management system (CDMS). Data will only be accessible to authorized personnel who require it to fulfill their duties within the scope of the research project. Participants are linked to a study identifier in the eCRFs and other project-specific documents. Any identifying information will be kept locked and only accessible to dedicated study team members at the clinic. The investigators will maintain appropriate medical and research records for this trial in compliance with the International Conference on Harmonization and Good Clinical Practice (ICH-GCP) [54] and regulatory and institutional requirements for the protection of the confidentiality of participants.

**Data security, access, and back-up.** Data is protected from unauthorized or accidental disclosure, alteration, deletion, copying, and theft. Traceability is ensured using REDCap and proper paper documentation. A role-based access system is implemented, with individual passwords assigned to each role (e.g., site investigator, statistician, monitor, administrator. All data entered by the eCRFs is transferred to the database using Transport Layer Security (TLS)

encryption. Each data point has attributes attached to it, identifying the user who entered it with the exact time and date. Retrospective alterations of data in the database are recorded in an audit table. The time, table, data field, altered value, and the user responsible are all recorded (audit trail). A multi-level backup system is implemented. Back-ups of the whole system, including the database, are run internally several times per day and on external tapes once a day.

**Analysis and archiving.** Health data collected during the study will be downloaded from the respective device and stored in a dedicated database for 10 years. Then, the data will be downloaded and stored locally for analysis and removed afterward.

**Electronic and central data validation.** The electronic data questionnaires mostly contain multiple-choice questions, which makes data validation easier. Data entry fields will be controlled to allow only adequate formats when possible.

## Statistical analysis

This study will evaluate the safety and efficacy of tirzepatide in conjunction with existing AID systems by assessing changes in TIR. It will also consider secondary and exploratory endpoints, including various CGM-based metrics, HbA1c levels, and surrogate markers for CVD, MAFLD, and renal function. All variables will be tested for normality using the Shapiro-Wilk and/or Kolmogorov-Smirnov tests. Continuous variables will be presented as mean±SD if normally distributed and median and interquartile ranges (IQR) if non-normally distributed, while categorical variables will be reported as N/%. Non-normally distributed variables will be analyzed using non-parametric tests. Statistical analyses will be performed using STATA® v.18.0 (StataCorp LLC, Texas, United States). and IBM® SPSS® Statistics version 30.0.0 (IBM Corp, New York, United States). Two data sets will be prepared: a per-protocol (PP) analysis set, including participants who complete treatment per protocol (from the end of titration until 16 weeks), and an intention-to-treat (ITT) analysis set, including all randomized participants as allocated initially. The ITT data set will be used for both safety and efficacy analyses. Due to the small sample size, no subgroup analysis is planned.

**Primary and secondary analysis.** The study design allows repeated measures (before and after) of the primary and exploratory outcomes. Repeated measures analysis of covariance (ANCOVA) will be used to assess the primary outcome, adjusting for baseline HbA1c, BMI, and TIR. Secondary outcomes will be assessed using a mixed-effect linear model for repeated measures (MMRM) from baseline through 8, 12, and 16 weeks of treatment, with baseline HbA1c and BMI as fixed effects and baseline TIR as a covariate. For both analyses, point estimates with 95% confidence interval and a two-sided p-value will be reported for the intervention (tirzepatide+AID vs. SoC+AID), with a 5% significance level. Model residuals will be inspected to verify approximate normality, and sensitivity analyses will be conducted to assess the robustness of findings under alternative model assumptions.

**Safety analysis.** The entire ITT population will be included for safety analysis (such as AE/SAE). A table will be populated with the frequency of system-wide adverse events between the two groups. Differences in safety outcomes will be examined between ITT groups using Chi-Square tests for categorical outcomes (any AE/SAE, clinically significance hypoglycemia, severe hypoglycemia, DKA) and non-parametric tests for continuous variables with non-normal distribution, e.g., CGM-measured time spent < 3.9 mmol/L.

Any deviations from the original statistical plan will be described and justified in the final report, as appropriate. Missing data will be addressed using a multiple imputation approach with 10 imputations for the intention-to-treat analysis [55].

## Monitoring

During the clinical study, monitoring visits will be conducted by authorized, qualified representatives of the Sponsor. The monitors will review all aspects of the study to ensure that the protocol and applicable regulatory requirements are adhered to and to assure participants' safety. Monitoring activities, as described in a study-specific monitoring plan, will include checking eCRFs for completeness and plausibility and verifying data against source documentation, reviewing the informed consent forms, checking the device accountability log, and ensuring that the investigator site files up to date

and contains all required documentation. All source data must be accessible to the monitoring personnel. Monitors must maintain patient confidentiality.

**Interim analysis and premature termination of the study.** As this is a 16-week clinical study, no interim analysis is planned. However, regular assessments will be conducted, and the trial will be prematurely terminated if more than three cases of severe hypoglycemia (level 3) or more than one case of DKA occur, provided these events are not attributed to the AID system malfunction. These criteria are based on exceeding the reported average incidence of severe hypoglycemia (11.8%) and DKA (4.8%) in patients with T1D, as per the T1D Exchange Clinic Registry [56]. The investigator must also discontinue the study intervention for an individual participant if any of the following criteria are met: withdrawal of informed consent, confirmed pregnancy, actively trying to become pregnant, developing an allergic reaction to tirzepatide, inclusion or exclusion criteria are no longer met, non-compliance with study procedures, or safety concerns at the investigator's discretion.

**Harms.** All AEs and SAEs occurring throughout the study (from participant consent until the completion of the last protocol-specific procedure, including the safety follow-up) will be documented and reported by the protocol. The time of onset, duration, resolution, action taken, intensity, relationship to the investigational product and study procedures, expectedness, and seriousness will be recorded in the eCRF.

SAEs must be reported to the study Sponsor within 24 hours. The Sponsor will reassess the SAE using REDCap. Fatal SAEs must be reported to the Competent Ethics Committee (CEC) within seven days. Suspected Unexpected Serious Adverse Reactions (SUSARs) will be reported to the CEC and Swissmedic (The Swiss Agency for Therapeutic Products or Competent Authority – CA) within seven days if fatal or 15 days for other events.

Foreseeable AEs such as mild hypoglycemia or device-related skin issues are not required to be reported unless they meet SAE criteria. Specific monitoring procedures are in place for hypersensitivity reactions, injection site reactions, cardiac conduction disorders, and severe gastrointestinal events. Any pregnancy occurring during treatment or within 30 days after tirzepatide discontinuation must be reported within 24 hours and followed to outcome. Female participants are required to report pregnancies occurring within this period.

For safety oversight, periodic safety reports (Annual Safety Reports, ASRs) will be submitted to regulatory authorities throughout the study. Follow-up procedures will ensure adequate medical care for all participants experiencing AEs, with continuous monitoring until resolution or stabilization.

**Audits and inspections.** Study sites, CA, and CEC have the right to perform inspections, and the sponsor has the right to conduct on-site auditing during working hours upon reasonable prior notice. The investigation documentation and source data/documents must be accessible to auditors/inspectors. Auditors/inspectors will maintain patient confidentiality.

**Confidentiality and data protection.** All data obtained in the context of the clinical study are subject to data protection. The investigator, their team, and the Sponsor must ensure the participants' privacy and protect their identities from unauthorized parties. On the eCRFs or other documents submitted to the Sponsor, participants must not be identified by their names but by a unique study identifier (pseudonymization). The investigator will maintain any study documents with subject names, e.g., subjects' written consent forms, in strict confidence and as part of the Investigator's Site File. Direct access to source documents will be permitted for monitoring, audits, and inspections. Only the study team will have access to the source data during the study.

The study data will be stored in a coded manner in Switzerland. In the frame of the study, data may be shared in a coded manner with a designated collaborator at the University of Antioquia in Medellín, Colombia, for statistical analysis purposes. Access to the coded data will be granted only to the designated collaborating researcher. The data will be transferred using a secure file-sharing platform that complies with Swiss and international data protection standards. This platform ensures end-to-end encryption to safeguard the data during transfer. The collaborating researcher is responsible for ensuring the secure storage and access of data on their end and must also use encrypted devices or

systems approved by their institution. Participants are informed of the possible data transfer and agree to it by consenting to participate in the study.

### Ethics

This study was already approved by the CA, *Swissmedic, d*as *Schweizerische Heilmittelinstitut* on 8 November 2024 and the *Ethikkommission Nordwest- und Zentralschweiz (EKNZ)* on 11 November 2024. Basec ID: 2024−01947.

### Protocol amendments

Substantial amendments will be only implemented after approval by the CEC and CA. Under emergency circumstances, deviations from the protocol may proceed without the authorities' approval (CEC and CA) to protect the participants' rights, safety, and well-being. Such deviations will be documented and reported to the Sponsor and the CEC/CA immediately (within two days).

All non-substantial amendments will be communicated to the CEC in the ASR and the CA as soon as possible. The ASR shall include any deviations from the protocol that may have affected the rights, safety, or well-being of the participants or the scientific integrity of the investigation.

### Publication and dissemination policy

The study results will be submitted for publication in internationally peer-reviewed scientific journals; members of the study team and collaborators will all be co-authors. The privacy and confidentiality of each participant shall be preserved in reports and data publication. Once results have been published, trial data will be accessible to external researchers. Investigators wishing to replicate the analyses or to do an individual patient meta-analysis may request the data from the corresponding author.

### Trial status and timeline

The current protocol is version 1.2, dated April 15, 2025. This trial was registered on ClinicalTrials.gov – U.S. National Library of Medicine – on October 18, 2024 (ClinicalTrials.gov ID: NCT06630585), and in the Swiss National Clinical Trial Portal (SNCTP ID: SNCTP000006174).

(i)   Recruitment Status: Patient recruitment began on 14 February 2025 and is currently ongoing. As of the submission date, three participants have been enrolled. Recruitment is expected to be completed by 31 January 2026.

(ii)  Data Collection: Data collection is projected to conclude in the third quarter (Q3) of 2026.

(iii) Results Availability: Study results are anticipated to be available between Q3 and Q4 of 2026.

## Discussion

This study aims to test the efficacy and safety of tirzepatide as an adjunct therapy to AID systems. We hypothesized that tirzepatide+AID is superior to AID alone in individuals with T1D on glucose control.

In the SURPASS trials, tirzepatide improved glycaemic control and body weight in adults with inadequately controlled T2D, including when used as a monotherapy, as an add-on to oral hypoglycemic agents, or as an add-on to titrated basal insulin. Reductions in HbA1c levels were seen as early as week four, nearing the maximum about week 24; fasting serum glucose (FSG) levels followed a similar temporal pattern. Treatment with tirzepatide also improved 7-point self-monitored blood glucose profiles relative to both placebo and active comparators and resulted in mean 2-h postprandial glucose levels that were generally within the normal range (≤ 140 mg/dL) at the study end [57]. Tirzepatide weight loss began by week four, did not plateau, was sustained for the duration of the trials, and was dose-dependent [57,58]. These findings

led to a significant increase in the number of patients achieving the target HbA1c proposed by diabetes clinical guidelines [59]. Regarding safety and tolerability, tirzepatide was generally well tolerated as monotherapy or add-on therapy to oral hypoglycemic agents or basal insulin in adults with T2D in nine SURPASS trials, with a safety profile consistent with that of selective GLP-1Ras [60]. In a head-to-head study that compares tirzepatide vs semaglutide (a potent long-acting GLP1-RA); tirzepatide was non-inferior and superior to semaglutide for the mean change in the HbA1c level from baseline to 40 weeks with favourable tolerability and safety [61].

In TD1 the clinical evidence of tirzepatide is still insufficient. A retrospective proof of concept conducted by Kaan Akturk, et al. [22] demonstrated that tirzepatide was associated with significant improvement in glycemic outcomes and reduction in weight without an increase in hypoglycemia or ketosis. Glycemic improvements were seen within the first three months. There was no clinically significant improvement in HbA1C beyond three months, despite continuous dose-dependent weight reduction. These data suggest that glycemic efficacy can be achieved in a short time frame with a 2.5 to 5 mg/week dose without further significant improvement with higher dosage and with adequate tolerability and safety. Consequently, randomized clinical trials are required to test the efficacy and safety of tirzepatide in T1D.

Furthermore, cardiovascular risk is a major concern in T1D. Epidemiological data indicate that individuals with T1D tend to experience CVD events, on average, more than a decade earlier than the general population [62]. Currently, no therapy for T1D is approved for both glycemic and cardiovascular indications. Tirzepatide has demonstrated a beneficial effect on other cardiometabolic risk markers, including lipid profiles and blood pressure, as well as improving liver fat content and reducing visceral and subcutaneous abdominal adipose tissue volume [30,34]. Moreover, the effect of tirzepatide on MACE is currently being evaluated in SURPASS-CVOT [63]. Since cardiovascular risk factors are increasing their prevalence among people with T1D, and CVD is the leading cause of death in people with T1D[62], exploring the effect of tirzepatide on cardio-renal outcomes, lipids, and biomarkers of MASLD is also the special importance.

This study has several strengths, including the use of a control group, baseline data collection during a run-in period, and the inclusion of a diverse range of AID systems. These strengths allow for a more robust comparison between groups, ensure a well-defined baseline for assessing changes over time, and enhance the generalizability of the findings.

One limitation of this study is its open-label design [64].In such trials, several types of bias may influence study outcomes; among the most notable are the Hawthorne effect, performance bias, and attrition bias [65,66]. To minimize the Hawthorne effect, both study arms will be observed equally, and a run-in period between baseline and randomization will allow participants to become accustomed to study procedures before randomization [67]. To reduce performance bias, all participants will follow a standardized protocol, with study staff trained to ensure consistent care and interactions across groups, and oversight by an independent monitor to confirm protocol adherence. In addition, the main study outcomes will be assessed using CGM sensors, which are not subject to investigator interpretation [68]. The risk of attrition bias will be mitigated by implementing active follow-up via reminders and phone calls, and with scheduled remote visits to minimize participant burden. An intention-to-treat approach will be applied to preserve the benefits of randomization and to mitigate the impact of drop-outs and loss to follow-up on the validity of the results [42,69]. Reasons for drop-out will also be documented to identify potential patterns [42]. These strategies aim to reduce, though not entirely eliminate, the potential for bias inherent in an open-label design..

Another limitation of our study is that the sample size and statistical power were calculated solely based on the primary outcome, TIR. As a result, the study may be underpowered to detect statistically significant differences in secondary or exploratory outcomes. These outcomes are designated as secondary because their primary purpose is to aid in the interpretation of the main outcome and to generate hypotheses that support future and larger research.This study aims to compare tirzepatide+AID with the standard of care (SoC) + AID in patients with T1D. It will provide unique data on the safety and efficacy of tirzepatide as an adjunct therapy to AID. The results could significantly improve glucose control and represent a paradigm shift in clinical practice, shifting the focus beyond insulin therapy.

## Supporting information

**S1 Table.** SPIRIT 2025 checklist: Recommended items to address in a clinical trial protocol.
(DOCX)

**S2 File.** Informed consent and participant information sheet.
(DOCX)

**S1 Text.** AID-JUNCT protocol.
(PDF)

## Acknowledgments

The authors thank the Diabetes Center of Bern (DCB), Bern, Switzerland, for their support in the regulatory processes and final submission to the authorities.

## Author contributions

**Conceptualization:** Maria Carolina Fragozo-Ramos, Matthias Hepprich, Jose Garcia-Tirado.

**Funding acquisition:** Jose Garcia-Tirado.

**Investigation:** Maria Carolina Fragozo-Ramos, Gabriela Schenker, Matthias Hepprich, Thomas Züger.

**Methodology:** Maria Carolina Fragozo-Ramos, Jaime Gallo-Villegas, Jose Garcia-Tirado.

**Project administration:** Maria Carolina Fragozo-Ramos.

**Supervision:** Thomas Züger, Jose Garcia-Tirado.

**Writing – original draft:** Maria Carolina Fragozo-Ramos.

**Writing – review & editing:** Maria Carolina Fragozo-Ramos, Gabriela Schenker, Matthias Hepprich, Jaime Gallo-Villegas, Thomas Züger, Jose Garcia-Tirado.

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
