## [Decision Letter · Decision Letter 0]

2 Jul 2025

Dear Dr. Garcia-Tirado,

Thank you for submitting your manuscript to PLOS ONE. After careful consideration, we feel that it has merit but does not fully meet PLOS ONE’s publication criteria as it currently stands. Therefore, we invite you to submit a revised version of the manuscript that addresses the points raised during the review process.

We look forward to receiving your revised manuscript.

Kind regards,

Timotius Ivan Hariyanto, M.D.

Academic Editor

PLOS ONE

**Journal Requirements:**

1. When submitting your revision, we need you to address these additional requirements. Please ensure that your manuscript meets PLOS ONE's style requirements, including those for file naming. The PLOS ONE style templates can be found at https://journals.plos.org/plosone/s/file?id=wjVg/PLOSOne_formatting_sample_main_body.pdf and https://journals.plos.org/plosone/s/file?id=ba62/PLOSOne_formatting_sample_title_authors_affiliations.pdf 2. We note that the grant information you provided in the ‘Funding Information’ and ‘Financial Disclosure’ sections do not match.  When you resubmit, please ensure that you provide the correct grant numbers for the awards you received for your study in the ‘Funding Information’ section. 3. In the online submission form, you indicated that “No datasets were generated or analyzed during the preparation of the current study protocol. Upon completion of the trial and publication of the results, de-identified individual participant data (including data dictionaries) will be made available upon reasonable request to the corresponding author. Access will be granted for research purposes, by applicable data protection regulations and institutional policies”.  All PLOS journals now require all data underlying the findings described in their manuscript to be freely available to other researchers, either a. In a public repository, b. Within the manuscript itself, or c. Uploaded as supplementary information.This policy applies to all data except where public deposition would breach compliance with the protocol approved by your research ethics board. If your data cannot be made publicly available for ethical or legal reasons (e.g., public availability would compromise patient privacy), please explain your reasons on resubmission and your exemption request will be escalated for approval. 4. When completing the data availability statement of the submission form, you indicated that you will make your data available on acceptance. We strongly recommend all authors decide on a data sharing plan before acceptance, as the process can be lengthy and hold up publication timelines. Please note that, though access restrictions are acceptable now, your entire data will need to be made freely accessible if your manuscript is accepted for publication. This policy applies to all data except where public deposition would breach compliance with the protocol approved by your research ethics board. If you are unable to adhere to our open data policy, please kindly revise your statement to explain your reasoning and we will seek the editor's input on an exemption. Please be assured that, once you have provided your new statement, the assessment of your exemption will not hold up the peer review process. 5. Your ethics statement should only appear in the Methods section of your manuscript. If your ethics statement is written in any section besides the Methods, please move it to the Methods section and delete it from any other section. Please ensure that your ethics statement is included in your manuscript, as the ethics statement entered into the online submission form will not be published alongside your manuscript.

Reviewers' comments:

Reviewer's Responses to Questions

**Comments to the Author**

1. Does the manuscript provide a valid rationale for the proposed study, with clearly identified and justified research questions?

Reviewer #1: Yes

Reviewer #2: Yes

2. Is the protocol technically sound and planned in a manner that will lead to a meaningful outcome and allow testing the stated hypotheses?

Reviewer #1: Yes

Reviewer #2: Yes

3. Is the methodology feasible and described in sufficient detail to allow the work to be replicable?

Reviewer #1: Yes

Reviewer #2: Yes

4. Have the authors described where all data underlying the findings will be made available when the study is complete?

Reviewer #1: Yes

Reviewer #2: Yes

5. Is the manuscript presented in an intelligible fashion and written in standard English?

Reviewer #1: Yes

Reviewer #2: Yes

You may also provide optional suggestions and comments to authors that they might find helpful in planning their study.

**Reviewer #1: ** The authors plan to recruit 42 participants in a prospective, randomized controlled trial design to evaluate the safety and efficacy of adding tripeptide to insulin therapy in T2D participants.

1. The sample size is calculated without considering the potential missing or dropout.

2. Please comment on how to avoid the potential bias from both participants and investigators as the open-label design is planned.

3. It’s unclear what imputation approach will be implement.

4. The ANCOVA and MMRM will be implemented with multiple covariates (e.g., hba1c, bmi, tir, group). With limited sample size, the robustness of the analysis results might be questionable.

5. Is there any plan to compare the baseline characteristics for two arms?

**Reviewer #2:**  Sample Size Justification:

1.The sample size calculation (n=42) assumes a 10.5% difference in time in range (TIR) between groups. Please provide references supporting this effect size estimate for tirzepatide in T1D, as most cited references appear to be from type 2 diabetes studies.

2.Consider whether this sample provides adequate power for secondary outcomes like hypoglycemia rates or cardiometabolic parameters.

Open-Label Design:

1.The potential for bias (performance, detection, and reporting bias) should be discussed in greater depth.

2.Consider adding blinded outcome adjudication for key endpoints like hypoglycemia and DKA events.

statistical Analysis:

1.The plan to use ANCOVA for primary analysis is appropriate, but please specify how missing data will be handled (e.g., multiple imputation methods).

2.Consider adding sensitivity analyses accounting for potential confounding by baseline characteristics.

**Do you want your identity to be public for this peer review?** For information about this choice, including consent withdrawal, please see our Privacy Policy

Reviewer #1: No

Reviewer #2: No

---

## [Author Response · Author response to Decision Letter 1]

1 Sep 2025

We sincerely thank the reviewers and the editor for their thoughtful and constructive feedback on our manuscript. A detailed point-by-point response has been provided in the uploaded rebuttal letter, and all suggested clarifications and revisions have been incorporated into the revised manuscript. We hope that these changes address the concerns raised and improve the overall clarity and quality of our work.

---

## [Decision Letter · Decision Letter 1]

7 Oct 2025

GIP/GLP-1RA as Adjunctive to Automated Insulin Delivery in Adults with Type 1 Diabetes (The AID-JUNCT Trial): Study Protocol for a Prospective, Randomized, Clinical Trial

PONE-D-25-24906R1

Dear Dr. Garcia-Tirado,

We’re pleased to inform you that your manuscript has been judged scientifically suitable for publication and will be formally accepted for publication once it meets all outstanding technical requirements.

Kind regards,

Timotius Ivan Hariyanto, M.D.

Academic Editor

PLOS ONE

Additional Editor Comments (optional):

Reviewers' comments:

Reviewer's Responses to Questions

**Comments to the Author**

1. Does the manuscript provide a valid rationale for the proposed study, with clearly identified and justified research questions?

Reviewer #1: Yes

2. Is the protocol technically sound and planned in a manner that will lead to a meaningful outcome and allow testing the stated hypotheses?

Reviewer #1: Yes

3. Is the methodology feasible and described in sufficient detail to allow the work to be replicable?

Reviewer #1: Yes

4. Have the authors described where all data underlying the findings will be made available when the study is complete?

Reviewer #1: No

5. Is the manuscript presented in an intelligible fashion and written in standard English?

Reviewer #1: Yes

You may also provide optional suggestions and comments to authors that they might find helpful in planning their study.

Reviewer #1: Thanks for addressing the raised comments satisfactory. This reviewer has no further concerns on the manuscript.

**Do you want your identity to be public for this peer review?** For information about this choice, including consent withdrawal, please see our Privacy Policy

Reviewer #1: No

---

## [Editor Report · Acceptance letter]

PONE-D-25-24906R1

PLOS ONE

Dear Dr. Garcia-Tirado,

I'm pleased to inform you that your manuscript has been deemed suitable for publication in PLOS ONE. Congratulations! Your manuscript is now being handed over to our production team.

Kind regards,

on behalf of

Dr. Timotius Ivan Hariyanto

Academic Editor

PLOS ONE